# TOWARD AGENTS THAT REASON ABOUT THEIR COMPUTATION

## ABSTRACT

While reinforcement learning agents can achieve superhuman performance in many complex tasks, they typically do not become more computationally efficient as they improve. In contrast, humans gradually require less cognitive effort as they become more proficient at a task. If agents could reason about their compute as they learn, could they similarly reduce their computation footprint? If they could, we could have more energy efficient agents or free up compute cycles for other processes like planning. In this paper, we experiment with showing agents the cost of their computation and giving them the ability to control when they use compute. We conduct our experiments on the Arcade Learning Environment, and our results demonstrate that with the same training compute budget, agents that reason about their compute perform better on 75% of games. Furthermore, these agents use 3 times less compute on average. We analyze individual games and show where agents gain these efficiencies.

## 1 INTRODUCTION

As reinforcement learning (RL) agents get better at a task, they typically do not become more energy efficient with experience. This is unlike humans, who adapt their cognitive process over time and can learn to devote cognitive effort with intention (Miller & Cohen, 2001). Agents, by contrast, often have their *computational processes* (a.k.a. *compute*) fixed; their processes for sensing, acting, learning and planning do not change after design time. So, when the problem demands more computation in some moments and less in others, the agent's computational process does not adapt.

There are many ways that agents are designed to use compute, and these choices have mostly been found by agent designers (e.g., Mnih et al., 2015; Silver et al., 2016; Vinyals et al., 2019; Bellemare et al., 2020; Wurman et al., 2022; Degrave et al., 2022; Mankowitz et al., 2023). Designers shape how and with what frequency the agent should use compute so that it can more effectively scale with available resources. As one simple example, the Arcade Learning Environment (Bellemare et al., 2013) (ALE) outputs 60 frames per second. Instead of making agents process each frame, making the agent process only every fifth frame as is the default with the ALE speeds up the execution by five times, with a negligible reduction in performance (Bellemare et al., 2012).

If agents could control their compute and received feedback on how they used it, they could begin to make these choices for themselves. But at present, designers choose how compute is used — designers decide when the agent should act or learn or plan, and the "cost" of compute is not made available to the agent. Additionally, agents are not typically long lived (Sutton et al., 2022), they are created to use compute in a certain way and replaced once their design limits them. This creates a *cycle of design*: where an agent is created, their performance is observed, the design is evaluated, and a new agent is improved that fits the trade-off known at design time. This cycle of design suggests that agents could do the same for themselves if they had access to similar actions and received feedback about their compute costs.

This work represents a foray into agents that can reason about their own computational processes. Since agents are capable of learning to make effective choices within the RL formalism, we will use the very same RL formalism to frame the challenge of reasoning about compute. We provide the agent with additional actions that do not have direct external effects, but rather change the agent's usage of compute. Additionally, we provide feedback that makes the cost of compute explicit alongside traditional environment reward. We focus on providing the agent with actions that change the rate

at which observations are processed and environment actions are computed, while modifying its rewards to reflect the cost of computation. We will show that in this setting agents can effectively learn to adapt their computational processes at run time from just their experience.

Our contributions are as follows:

1. We show that an agent with the ability to control its compute and see the consequences of its computation can learn to adapt its compute at run time.

2. These agents perform better on 75% of the Arcade Learning Environment games, while using the same training compute budget. They learn compute efficient behaviors per-game that are on average 3 times more efficient. We show that efficiencies are learned and tailored to each game, and adapt to the changing circumstances within a game.

3. We show that agents are responsive to the experienced compute cost, showing that an agent's computational processes adapt to the RL task posed to it by the designer.

## 2 REASONING ABOUT COMPUTATION

In this section, we show that we can use the RL problem formulation to enable agents to reason about their compute. Our control problem consists of observations and actions. The agent processes an observation into a state, $S_t$, at a time step, $t$, and computes an action, $A_t$, in the same time step. This action is then sent to the environment, and in response, a new observation is perceived in the next time step resulting in the next state $S_{t+1}$. A special part of the agent's observation is a scalar reward signal, $R_{t+1}$, received at every time step. This cycle of interaction repeats in a synchronized manner, and the objective of the RL agent is to maximize the expected episodic discounted return: $G = \mathbb{E}\left[\sum_{k=1}^{T-1} \gamma^{k-1} R_{k+1}\right]$.

### 2.1 FRAMING THE OBJECTIVE

The agent's goals are specified in the reward signal (Sutton, 2004; Bowling et al., 2023), and so, the scalar reward should also provide the consequences of the agent's computational processes. For instance, consider how an agent may learn when it is important to act while operating a solar panel tracking station. The solar panel needs to be continually oriented toward the sun to capture sunlight throughout the day, where a better agent is one that maximizes the power exported to the grid. The agent can learn to track the sun accurately by actuating the panel and observing the exported energy change in value. However, the agent's computation consumes energy as well, which is energy not exported to the grid. If the objective incorporated the energy that has been gathered, $r_t^{\text{task}}$, and the compute cost of the agent, $c_t$, a scalar reward can be constructed representing the total power exported to the grid: $r_t = r_t^{\text{task}} - c_t$. This frames the problem the designers face — balancing the agent's compute use with performance on the task, but now allows the agent to see feedback on its compute usage. In the next section, we show how actions that change the cost can enable the agent to choose when to use compute.

### 2.2 CONTROLLING THE COMPUTATIONAL PROCESSES

The rate at which agents process observations and take action greatly changes their compute cost. If the agent processes at half the rate, the agent can halve its compute cost. This is explicitly true for all agents with fixed rates of environment interactions with which they store experience, learn from experience, or generate experience from a model for training or choosing actions. When an agent takes an action, it could also decide how long it should keep repeating that action until it processes another observation. This gives the agent direct control over the rate at which it processes observations and takes actions, and therefore control over its compute usage. The options framework (Sutton et al., 1999) abstractly defines temporally extended courses of action, and so, we use options to model our expanded concept of action to let the agent control when it will process the next observation.

We provide the agent with a set of options $\mathcal{O} = \mathcal{A} \times \mathcal{T}$, where $\mathcal{T} \subset \{1, 2, \ldots\}$ is a set of possible durations an action can be repeated. We assume the set of durations are given but these could be discovered by the agent itself (Harb et al., 2018; Barreto et al., 2019). While an option is being executed the agent uses a negligible amount of compute. However, the selection of the option to

execute is compute intensive, invoking all of the agent's processes for constructing state, producing an action, storing experience, learning, planning, etc. We tie computation cost $c_t$ at time $t$ directly to whether an option was selected,

$$c_{t+1} \overset{\text{def}}{=} \begin{cases} c & \text{if an option is selected, at time } t, \text{ or} \\ 0 & \text{during option execution.} \end{cases} \tag{1}$$

The agent's policy $\pi : \mathcal{S} \to \Delta(\mathcal{O})$, maps the current state $s_t$ to a distribution over the set of options, for which $o_t$ is sampled. We will focus on value-based RL methods that select an option using an option-value function $Q : \mathcal{S} \times \mathcal{O} \to \mathbb{R}$, which are learned from experience. The option-value function can be updated as in a semi-MDP (Sutton et al., 1999), which due to the fixed option duration is equivalent to employing a $\tau$-step TD-update for option $o_t = \langle a_t, \tau_t \rangle$. The TD error for selecting this option at time $t$ is then:

$$\delta_t = \left( (r_{t+1} - c) + \gamma r_{t+2} + \ldots + \gamma^{\tau-1} r_{t+\tau} + \gamma^{\tau} \max_o Q\left(s_{t+\tau}, o\right) - Q(s_t, o_t) \right). \tag{2}$$

Note the introduction of the cost of compute $c$ which is incurred at time $t + 1$ after the option was selected while the compute cost $c_{t+t} = \ldots = c_{t+\tau} = 0$ and is dropped.

We have provided the agent with a means to control its compute usage through the selection of $\tau_t$ in option $o_t$, while also providing it with feedback about the cost of compute through $c_t$. By providing a variety of option durations $\mathcal{T}$, the agent can learn when precisely timed control is necessary (choosing shorter options), and when compute should be conserved (choosing longer options). We can now explore the central question of the paper: Can an agent with the ability to control its compute and see the cost of its computation learn to adapt its compute usage at run time.

## 3 EXPERIMENTAL SETUP

We evaluate on the Arcade Learning Environment (ALE) (Bellemare et al., 2013; Machado et al., 2018a), which provides many diverse, stochastic games. Agents use sticky actions with the default probability and a frame-skip of 5. Our baseline is Deep Q-Network (DQN) (Mnih et al., 2015), with standard modern improvements: Adam optimization (Kingma & Ba, 2015), activation normalization after each convolutional layer (Ioffe & Szegedy, 2015; Zhang & Sennrich, 2019), and gradient clipping at magnitude 1 (Schwarzer et al., 2023; Clark et al., 2024). These changes improve stability while preserving the core DQN design.

We measure compute by the number of *decisions*, $d$, where each decision invokes the agent's full process for constructing state, producing an action, storing experience, and learning. In our experiments, we will often refer to the number of decisions an agent has made $d_T = \sum_{t=1}^T \mathbb{1}(c_t \neq 0)$ or the agent's decision rate $d_T/T$, which for ease of presentation we present as decisions per second (Hz), or $60 d_T/T$, as all of our experiments are in Atari games that operate at 60 frames per second. Standard DQN acts every 5 frames, which fixes its decision rate at 12 Hz. Agents that choose to make decisions less frequently, reducing their compute usage, would see a decisions per second less than 12 Hz. We deliberately focus on counting decisions rather than wall-clock time, or watt-hours as it is hardware and software independent and is directly correlated with floating-point operations, providing a stable basis for comparison.

Our approach, *Compute DQN*, extends DQN by adding actions that control decision frequency. We generate options using the Cartesian product of the reduced action set per-game and durations $\mathcal{T} = \{1, 2, 4, 8\}$: $\mathcal{O} = \mathcal{A} \times \mathcal{T}$. This lets the agent operate at DQN's fixed rate of 12 Hz (every 5 frames) or 6 Hz, 3 Hz and down to 1.5 Hz (i.e., acting as rarely as every 40 frames), giving coarse control over its decision rate. Other than that, Compute DQN operates almost identically to DQN treating its options as actions. Compute DQN *reasons* about compute by explicitly representing the expected value of different compute-actions that each impact the return under our explicit compute-performance trade-off. It learns a state-option value function, updating after a fixed number of decisions, by training on tuples of experience from its replay buffer, $\langle s_t, o_t, r_{t:t+\tau} - c, s_{t+\tau} \rangle$, where $r_{t:t+\tau} = \sum_{i=1}^{\tau} \gamma^{i-1} r_{t+i}$ is the discounted sum of rewards received over the duration of the option. Additionally, in calculating its TD-update it discounts the value of the stored next state from its target network by $\gamma^{\tau}$, to account for the uneven option durations as in Equation 2.

Both agents are trained with the same compute budget. We fix training to 40 million decisions, which matches the compute used by DQN when trained on 200 million frames with a frame skip of 5.

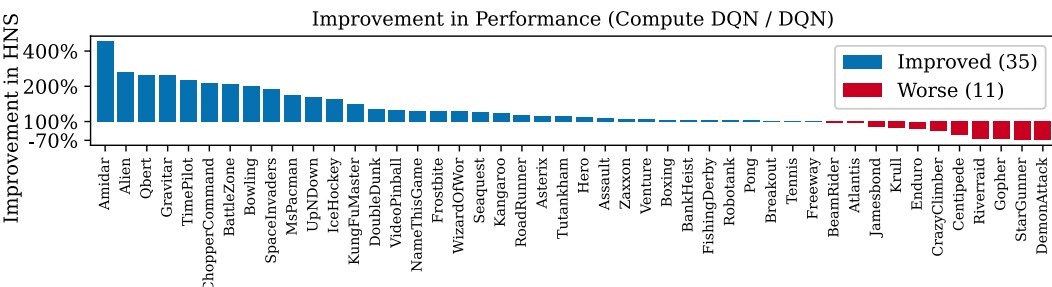

Figure 1: Improvement of Compute DQN over DQN in human normalized score (HNS) across 46 Atari games. Compute DQN achieves higher HNS in 75% of games. Amidar shows the largest gain of 487%, while the largest loss is 30% below DQN.

Keeping the number of decisions constant lets us focus on compute efficiency over sample efficiency. Indeed, agents that reduce their compute usage through their decision rate may use more emulator steps and complete more episodes within a training run. However, all agents receive the same total tuples of experience and do the same total learning updates.

The order of magnitude of what constitutes a good score varies considerably between games. Since we want the agent to face an interesting tradeoff between its compute usage and its performance we chose a different cost of computation $c$ for each game. We chose $c = \frac{G_T}{T}$, where $G_T$ is the accumulated undiscounted return[1] achieved by DQN. In an actual application, we would expect it to be more natural to set $c$ by an actual tradeoff between the agent achieving its goals and the cost of its compute to pursue those goals. However, this approach of tying the compute cost to the agent's average reward could give a knob to emphasize the importance of compute efficiency by doubling or halving this calculated $c$. All results are averaged over 10 independent runs with different seeds for network initialization and environment stochasticity[2].

## 4 RESULTS

We first present overall results across the Arcade Learning Environment's suite of games, then analyze three games in detail. Our analysis shows that agents effectively learn to efficiently use compute from experience: their strategies improve with experience, are tailored to each game, and adapt to the changing circumstances within a game. We further show that agents can adapt to different costs of compute, using more or less compute as $c$ is varied. These results show that agents not only learn to use compute from experience, but also adjust their decision rates consistently across games, game specific moments, and costs.

In all of our results we will focus on two measures: performance and compute usage. For performance we report the undiscounted return averaged over the last 100 training episodes across 10 seeds, using game score as reward.[3] We then normalize this score to produce a Human Normalized Score (HNS) (Mnih et al., 2015), allowing us to more fairly compare across games. For compute use we report the decisions per second (Hz) rate of the agent relative to the 60 frames per second real-time rate of Atari.

### 4.1 PERFORMANCE AND COMPUTE USE

Under equal training compute, Compute DQN outperforms DQN on 75% of Atari games (see Figure 1). The largest gain is on Amidar with 487%, while the largest loss is on DemonAttack with

---

[1]The return $G_T$ is based on clipped rewards. As is typical with the ALE, agent rewards are clipped to the range $[-1, 1]$ to handle the high variation in rewards scales across games. As such, the cost $c$ is also guaranteed to be in the range $[-1, 1]$.

[2]Github repository will be made available for the published version

[3]We removed three games — Asteroids, PrivateEye, and MontezumaRevenge — as DQN achieved less than 10% HNS making relative improvements uninformative. Nevertheless, Compute DQN was observed to outperform DQN on MontezumaRevenge and Asteroids and was worse on PrivateEye.

30% below DQN (see Figure 9 in the Appendix for raw HNS of the algorithms). Note that the goal was to give actions to allow the agent to balance the tradeoff between performance and compute, not improve performance. It may seem odd that there would be a performance improvement at all. However, this result is consistent with others who have suggested repeated actions improve credit assignment (McGovern & Sutton, 2005) possibly the result of using an n-step update; or repeat actions do better exploration (Dabney et al., 2021), or are more sample efficient (Braylan et al., 2015). Other causes may also include Compute DQN seeing a larger number of total frames for the same training compute budget, or the replay buffer containing a larger variety of states over a larger number of episodes, although as we will see in analyzing the agent's compute usage, reduced decision rates do not seem to correlate with the performance improvement. In summary, these results show that adding actions to control compute need not reduce the agent's ability to perform the task.

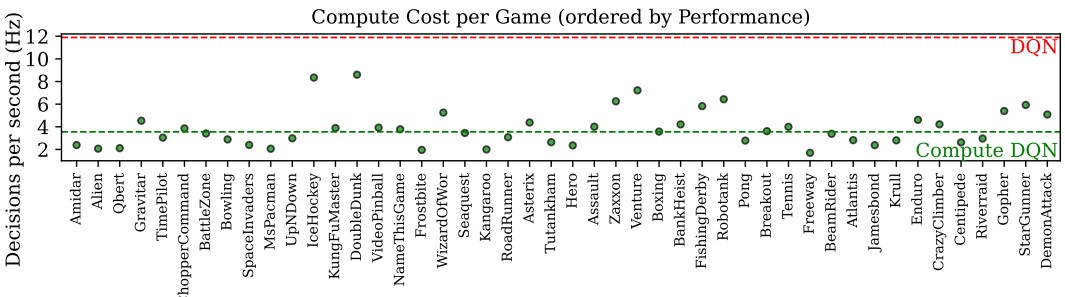

Figure 2: Compute DQN learns game-specific decision rates across the Arcade suite. Each green dot shows the average decisions per second for a single game, measured over 10 independently trained agents. The dashed green line indicates the suite-wide mean of 3.6 Hz. For comparison, Compute DQN uses 3.4 times less compute than DQN, which acts at a fixed 12 Hz (red dashed line). These results demonstrate that Compute DQN can adapt its compute use per game while maintaining performance.

Compute DQN's primary goal is to reduce compute usage, for which it is also very effective (see Figure 2). On average it uses 3.4 times fewer decisions per second than DQN (3.6 Hz vs. 12 Hz). Decision rates vary across games, including those with the largest HNS gains and losses, showing that reduced compute does not necessitate reduced performance in a game. Furthermore, the lack of correlation between reduced compute usage and performance gives evidence that performance improvements are not driven by simply a larger number of emulator steps being used during training (See Table 1 in the Appendix for enumator steps). While human action rates are not available for the ALE, StarCraft II professionals average ∼174 actions/min (Vinyals et al., 2019); Compute DQN's learned average of 3.6 Hz (≈216 actions/min) is closer to this human scale than DQN's fixed 720 actions/min, suggesting that compute is used deliberately. These aggregate results show that agents adapt their decision rates across games without hurting performance, raising the question of how this adaptation is achieved. We explore this next by taking Castro's (2024) advice and narrow in on a few games rather than performance in aggregate.

### 4.2 EFFICIENT USAGE OF COMPUTE IS LEARNED FROM EXPERIENCE

Figure 3 shows how decision rates evolve alongside performance during training. First, we can see that agents indeed learn to reduce their compute with gained experienced. The exact trend, though, varies between the games. In Pong, performance improves considerably early in learning without a distinct reduction in the decision rate. Only after the policy is quite strong do we see the agent favoring lower decision rates, finding action sequences that still preserve high scores while reducing the computation cost. This is in contrast to Breakout and Asterix, where both see a sharp drop in the decision rate early in learning. Before good sequences of actions are found, the optimal choice is to simply make fewer decisions and avoid the computation cost. This is most clear in Breakout where the decision rate is halved in the first million decisions. As the agent's policy improves in Breakout and Asterix, though, the tradeoff reverses as the agent begins to return to taking more frequent actions that now lead to considerable increases in performance.

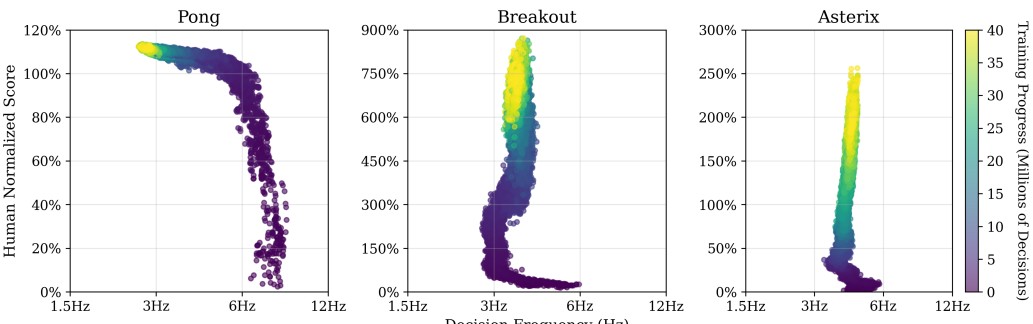

Figure 3: Population dynamics of learning: each point is a moment in time during training, reported across 10 seeds per game. Color encodes training progress.

### 4.3 INVESTIGATING THE LEARNED STRATEGIES

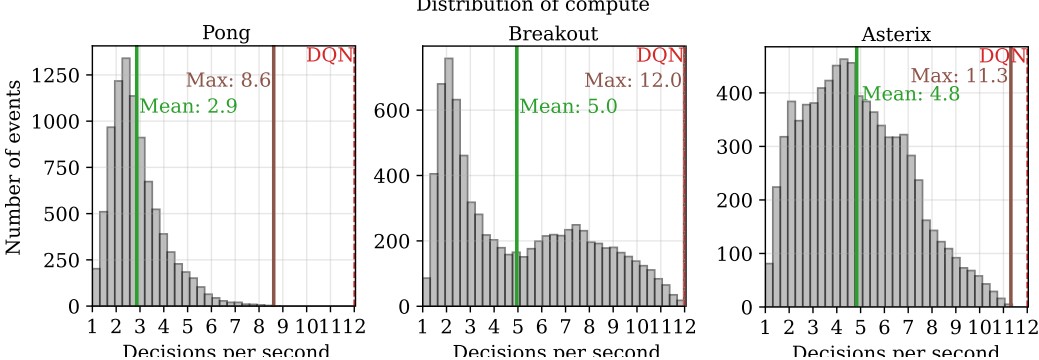

Figure 4: The decision rate within a trajectory differs by game. Pong and Asterix are uni modal, whereas Breakout is bimodal.

We now examine a single agent trained to completion for each game and consider its learned policy. Figure 4 shows the distribution of the decision rate over a complete trajectory. The agent's learned decision rate is different across games and show distinct game-specific patterns. In Pong, decision rates cluster tightly around the mean, with only occasional spikes. Breakout shows a bimodal distribution, indicating that the game may alternate between moments that are less intensive and others that require frequent decision making. Asterix shows a more broad, uniform distribution.

We further analyze these trajectories individually in the next section. To let readers form their own view first, we provide videos showing the game, and the agent's decision rate[4].

#### 4.3.1 PONG

In Pong, the agent reduces compute when acting would not get the agent in a better position to strike the ball. Instead, it tends to make more frequent actions moments before hitting the ball (in red), or to reposition. Additionally, we see general trends of conserving compute, where the decision rate drops after the player hits the ball (in green), and when the ball is heading toward the opponent (in blue).

#### 4.3.2 BREAKOUT

The difficulty of Breakout increases in well-defined stages: the ball accelerates after consecutive blocks have been hit, and striking the back wall both increases speed and shrinks the paddle (Bre,

---

[4]Videos on the three discussed games along with others are in the supplementary material and will be uploaded for the camera ready version.

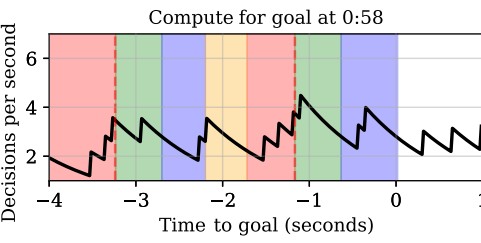 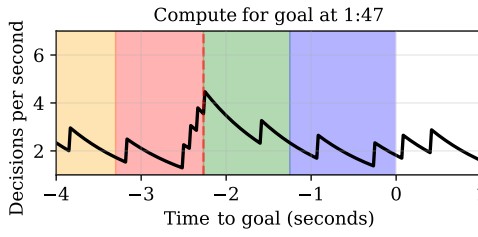

(a) Two hits: one fast, the other is slow.      (b) A slow ball hit by the agent.

Figure 5: Exponentially decayed average of decision rate for Compute DQN on segments of an episode in Pong. Coloured regions indicate when the opponent has just hit the ball, the ball is approaching the player and dotted line indicates when the ball is hit, the player has just hit the ball, and the ball is heading toward the opponent. There are typically rises in decision rate in the red region in anticipation of the ball reaching the player's paddle.

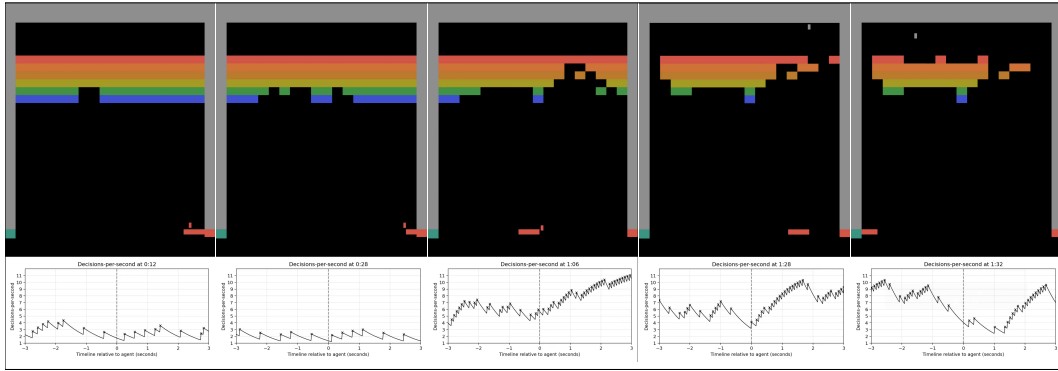

Figure 6: A sequence of observations (top) with an associated graph of the exponentially decayed average of decision rate at that moment (bottom) for Compute DQN in Breakout. As the game progresses the decision rate increases considerably, in correlation with the ball speeding up as blocks are cleared and a "breakout" occurs. The final image shows an example decision rate dropping while the ball is behind the wall of blocks.

2025) These mechanics gradually increase the precision of control needed to continue to keep the ball in play and score points. In Figure 6, we see that the first two images show moments when the ball speed increases by a small amount, and the agent does not increases its use of compute. Whereas the next two images show two moments: the first where the ball further increases in speed, and the next where the paddle width decreases, both resulting in increases in the decision rate. We see in the final image that the agent conserves compute when the ball is stuck behind the wall and no immediate action is needed.

### 4.3.3 ASTERIX

Asterix is a game where the agent needs to collect various ancient objects (knights, shields, lamps), and the player needs to avoid the lyre. As the game progresses, the objects change, and after collecting fifty of one object, the next group of objects appears moving faster than the previous objects. The agent concentrates compute during dense collectible waves and lowers it between waves (Figure 7), and as objects speed up the decision rate increases proportionally.

### 4.4 AGENTS BALANCE COMPUTE COSTS

As a final experiment, we vary the per-decision compute cost, $c$, and observe how these compute costs change the learned policies. For each cost, we ran 10 seeds for 10 million decisions. In Figure 8, the moderate setting matches the DQN-derived cost; we also test higher costs $10c$, $5c$, and lower costs $\frac{c}{5}$, $\frac{c}{10}$. We see that as the cost increases, policies choose longer options to reduce their decision

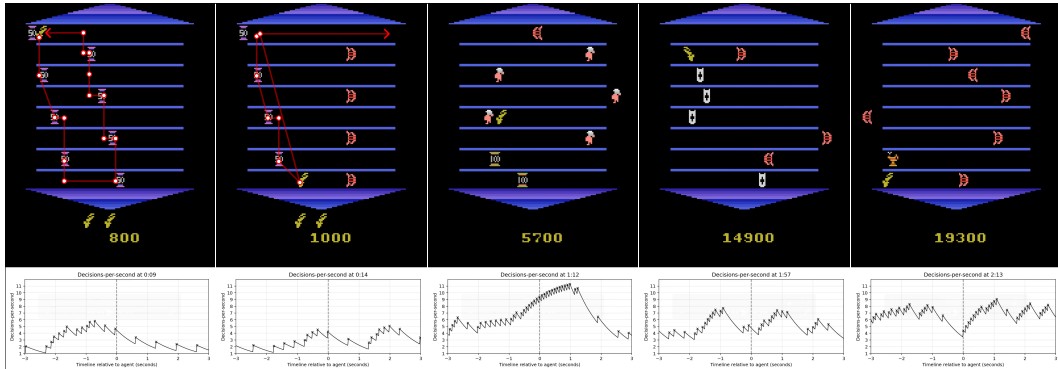

Figure 7: A sequence of observations (top) with an associated graph of the exponentially decayed average of decision rate at that moment (bottom) for Compute DQN in Asterix. The decision rate grows as the waves get faster and have more collectibles. In the first two images, the agent has just finished collecting the objects in a wave, resulting in a spike in decisions followed by a quick decrease awaiting the next wave. The remaining images repeat this structure with very high decision rates as the number and speed of objects rises with dips between waves.

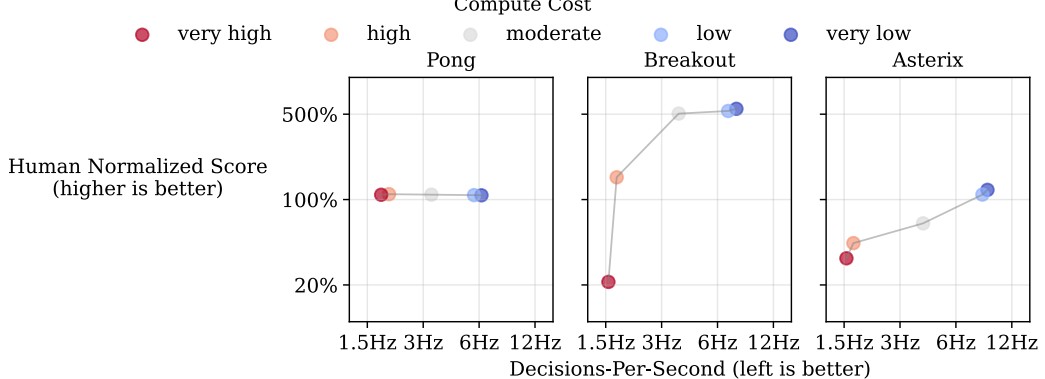

Figure 8: Varying the compute cost, $c$, changes the learned decision rate and performance. Higher costs yield fewer decisions per-second; low costs increase the decision rate. Each game shows a performance-compute frontier, showing that agents adapt to their cost parameter.

rate; as the cost approaches zero, policies operate closer to 12 Hz. In Pong, the Pareto tradeoff curve is flat suggesting that a strong policy still exists even at very low decision rates. We suspect that moderate to low compute costs might provide a small gradient for agents to decrease their decision rate, even though a large reduction in decision rate after 10 million decisions is possible after much more training time (referring back to Figure 3). Under higher compute costs, reducing the decision rate is likely prioritized much earlier, which still gives the agent time to learn this compute efficient, yet strong, policy. In Breakout and Asterix we see a more traditional Pareto curve where performance is sacrificed for compute efficiency under high compute costs. These results show that agents are respecting the compute-performance tradeoff across high and low cost regimes, while seeing monotonic increases in the use of compute as it becomes cheaper.

## 5 DISCUSSION & RELATED WORK

Several works have considered ideas related to repeating actions. Some show that action repeats can improve credit assignment (McGovern & Sutton, 2005) and sample efficiency (Bellemare et al., 2012; Mnih et al., 2015; Braylan et al., 2015), or aid exploration (Dabney et al., 2021). Others allow agents to dynamically select the action repeat length (Lakshminarayanan et al., 2016; Sharma et al., 2017; Vinyals et al., 2019; Biedenkapp et al., 2021), similar to our approach. However, these studies are

focused on improvements to sample efficiency rather than compute efficiency. They do not consider a cost for compute, nor how an agent adapts when reducing compute is part of the objective. A similar approach to ours was considered for continuous-time tasks with a fixed known horizon (Treven et al., 2024), where they reduce their problem to a discrete time RL problem (possibly including an interaction cost). They investigate problems with relatively short time horizons compared to Atari as their solution involves learning policies with time-remaining as an input.

Our work is placed within the long standing field of conditional computation in AI. In neural networks, early work of Bengio et al. (2013) and Davis & Arel (2014) compare several ways to backpropagate through a stochastic neuron, showing that a neural network can learn to conditionally activate costly parts of the model. Our approach of letting the agent choose its decision rate could be viewed as delaying the rate at which the whole network is activated. Since, the field has developed (Bengio et al., 2015; Bolukbasi et al., 2017; Graves, 2017; Banino et al., 2021) showing regularisers that encourage a trade-off between prediction accuracy and compute cost, allowing the neural network to learn when to halt a recurrent computation and output an answer. In a parallel effort, the cognitive science field proposes agents that reason about their computations (Hay et al., 2012; Callaway et al., 2018) as a form of meta-learning. And more recently, RL methods could produce intermediate outputs (Hanna & Corrado, 2025) reminiscent of chain-of-thought reasoning in LLMs (Wei et al., 2022), where notions of using compute efficiently are gaining further attention (e.g., Achille & Soatto, 2025; Raposo et al., 2024). These works take a similar approach in that internal actions that modify an agent's computational processes can be treated like any other action. However, they do not show such agents learning to be more compute efficient or learning to use their compute deliberately, such as our experiments on the Arcade Learning Environment. Finally, the AlphaStar agent (Vinyals et al., 2019) imposed a decision-rate limit, but the focus of their work was to enforce fair play with a human opponent.

We have shown that agents with access to actions that make trade-offs between performance and compute can reason about the impact of these actions on their expected return. These agents reason by explicitly representing the expected value of selecting a compute-action and how it impacts the return under our explicit compute-performance trade-off. We have shown that giving agents options that delay computation can balance compute costs with performance, however, options are not limited to action repeats, options may be arbitrary sequences of actions; performed open or closed loop. Options can also be stochastic or contingent on events (Sutton et al., 1999). In the longer term, agents could autonomously discover options (Machado et al., 2018b; Barreto et al., 2019; Harb et al., 2018) with fast execution times that might allow for even lower decision rates. Adding more available options may increase the complexity of the task for the agent, but with more approximate methods, agents could learn about many options from fragments of experience consistent with executing other options (Sutton et al., 1998), or plan over them with model-based methods (Sutton, 1991; Huang et al., 2025).

Agents today are typically not long lived (Sutton et al., 2022): they are trained after which learning is stopped before possible deployment as a fixed policy. One challenge for agents that continually learn is the big world hypothesis (Javed & Sutton, 2024): the world is large and complex, and designers cannot anticipate how the design of an agent might need to change over time. If the agent's computational processes are fixed, it may fail to utilize available resources efficiently as the agent experiences new parts of the "big world". Furthermore, a long-lived agent may gain access to more and cheaper compute resources throughout its lifetime; possibly reflected in lower compute cost. Such a long-lived agent will need to remain adaptive not just to a constantly changing world, but a constantly availability of resources, or a changing tradeoff of compute costs.

There are many untapped directions for designers to create agents that can reason about their compute. We hope that future work will investigate providing richer compute-actions beyond controlling the decision rate. And one would need to be careful that the computational cost of executing these compute-actions is (i) accounted for if not negligible and (ii) much smaller than the computation cost of the agent's typical action selection.

## 6 CONCLUSION

In this paper we presented Compute DQN, a stepping stone toward agents that can reason about their own computational processes just as they reason about how they can bring about goals in the

world. Our approach aimed to unify the goals of the agent with that of the agent designer, by giving the agent feedback on its computation cost, and providing the agent actions to control how it uses compute by adapting its decision rate. We showed that with these additions, Compute DQN learns interesting and deliberate strategies for using compute that improve with experience, are tailored to its environment, and adapt to its current situation. We showed performance improvements with the same training budget on 75% of ALE games, while also showing a 3 times reduction in compute usage. We believe this work opens up future directions for agent's to automatically adapt other aspects of their computational processes. For example, they might be able to adaptively select when they can process observations with a larger or smaller neural network; how frequently to sample experience from a model; or adapting batch size or replay ratio (Fedus et al., 2020). Furthermore, there is an untapped direction in finding more sample efficient means to train agents that reason about their compute, such directions could include reward shaping to anneal the cost over time. In addition, intra-option learning and model-based methods could help mitigate the additional learning burden created when giving agents more control over their compute usage. Agents that can flexibly adapt their compute requirements present opportunities for more energy efficient agents, cheaper and faster RL experiments, more powerful agents that optimally make use of available compute, and long-lived agents that can continually adjust to compute needs beyond what can be designed in advance.

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

# A  APPENDIX

## A.1  RESULTS

We present the human normalized score to compare both methods across games in Figure 9. By computing the geometric average over human normalised scores, we see that across the ALE, DQN averages 188.58% HNS, and Compute DQN averages a higher percentage of 232.89%. As it is more common to present the number of games improved upon, we report the 75% of games improved throughout the main text.

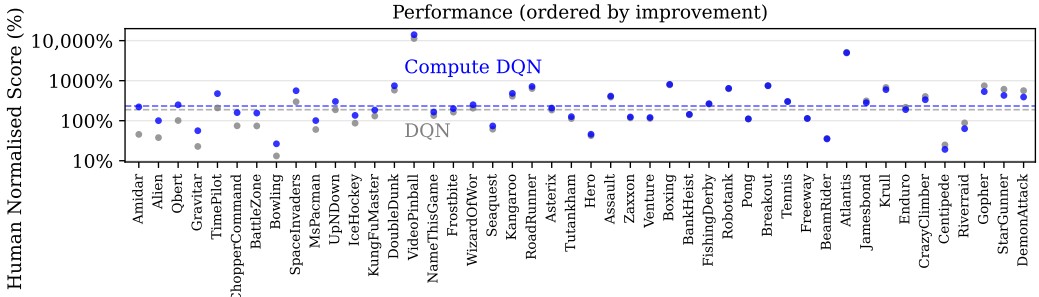

Figure 9: We present the human normalized score to validate that (i) DQN and Compute DQN on average perform better than a human (above 100%), and (ii) the performance loss (seen on the right) from Compute DQN is marginal relative to the performance benefit (seen on the left).

We report the number of decisions, updates, and frames emitted by the emulator in Table 1. Compute DQN uses the same amount of training compute as DQN by keeping the decisions and updates equal. However, as Compute DQN's mechanism to save compute is to skip frames, this causes the emulator to produce more frames. We observe that the reduced decision rates seen per-game do not seem to correlate with the performance improvements in that game, and so the performance benefit appears to be more largely influenced by the game itself.

| Environment | Decisions (M) | Updates (M) | Emulator Steps (M) |
|---|---|---|---|
| Alien | 40 | 10 | 1030 |
| Amidar | 40 | 10 | 806 |
| Assault | 40 | 10 | 590 |
| Asterix | 40 | 10 | 571 |
| Atlantis | 40 | 10 | 822 |
| BankHeist | 40 | 10 | 592 |
| BattleZone | 40 | 10 | 594 |
| BeamRider | 40 | 10 | 580 |
| Bowling | 40 | 10 | 637 |
| Boxing | 40 | 10 | 557 |
| Breakout | 40 | 10 | 638 |
| Centipede | 40 | 10 | 815 |
| ChopperCommand | 40 | 10 | 543 |
| CrazyClimber | 40 | 10 | 562 |
| DemonAttack | 40 | 10 | 464 |
| DoubleDunk | 40 | 10 | 262 |
| Enduro | 40 | 10 | 462 |
| FishingDerby | 40 | 10 | 359 |
| Freeway | 40 | 10 | 1341 |
| Frostbite | 40 | 10 | 1008 |
| Gopher | 40 | 10 | 422 |
| Gravitar | 40 | 10 | 440 |
| Hero | 40 | 10 | 859 |
| IceHockey | 40 | 10 | 280 |
| Jamesbond | 40 | 10 | 810 |
| Kangaroo | 40 | 10 | 972 |
| Krull | 40 | 10 | 846 |
| KungFuMaster | 40 | 10 | 566 |
| MsPacman | 40 | 10 | 1034 |
| NameThisGame | 40 | 10 | 557 |
| Pong | 40 | 10 | 710 |
| Qbert | 40 | 10 | 1018 |
| Riverraid | 40 | 10 | 748 |
| RoadRunner | 40 | 10 | 710 |
| Robotank | 40 | 10 | 351 |
| Seaquest | 40 | 10 | 768 |
| SpaceInvaders | 40 | 10 | 942 |
| StarGunner | 40 | 10 | 530 |
| Tennis | 40 | 10 | 452 |
| TimePilot | 40 | 10 | 730 |
| Tutankham | 40 | 10 | 904 |
| UpNDown | 40 | 10 | 704 |
| Venture | 40 | 10 | 335 |
| VideoPinball | 40 | 10 | 814 |
| WizardOfWor | 40 | 10 | 410 |
| Zaxxon | 40 | 10 | 337 |

Table 1: Compute DQN is trained with the same number of decisions and updates as DQN (40M decisions, 10M updates), however, as Compute DQN skips additional frames to reduce its compute cost, the emulator steps more for more efficient agents.

### A.1.1    TRAINING DQN USING THE AVERAGE DECISION RATE LEARNED BY COMPUTE DQN

We provide an additional results to show that the performance improvement at the lower decision rate cannot be achieved by an agent designed with a fixed decision rate of 4 Hz. We chose 4 Hz over exactly matching 3.6 Hz to provide DQN a slightly higher decision rate, as our investigation

here is primarily focused on analysing the change in performance. If DQN performs higher at 4 Hz, this indicates that there exists strategies to play each game such that DQN could learn to play very efficiently. Our results seen in Figure 11 show that Compute DQN is still able to perform better on 61% of games.

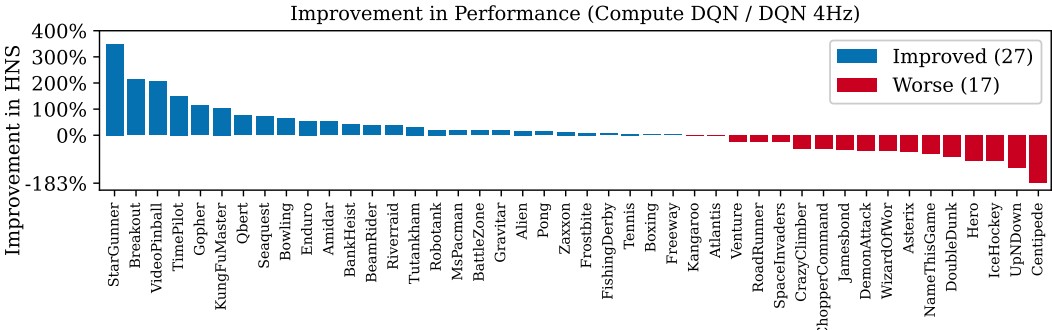

Figure 10: Improvement of Compute DQN over DQN (at 4 Hz) in human normalized score (HNS) across 46 Atari games. Compute DQN achieves higher HNS in 61% of games. StarGunner shows the largest gain of 349%, while the largest loss of 183% below DQN.

We present the human normalized score to compare both methods across games in Figure 11. By computing the geometric average over human normalised scores, we see that DQN (4 Hz) averages 202% HNS across the ALE, and Compute DQN averages a higher percentage of 225%. As it is more common to present the number of games improved upon, we see that Compute DQN is better on 61% of games whilst achieving the same compute efficiency as DQN fixed to 4 Hz.

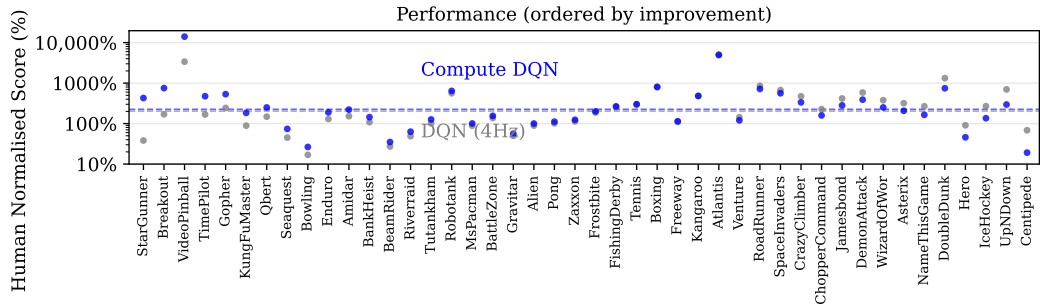

Figure 11: We present the human normalized score to validate that (i) DQN (4 Hz) and Compute DQN on average perform better than a human (above 100%). The performance gains for StarGunner and Breakout (both left), and the performance loss on Centipede are large (right). On average, we see that giving agents control over their compute still results in a performance and efficiency improvement averaged across the suite.

