# OpenReview forum: "Toward Agents That Reason About Their Computation"
_ICLR.cc/2026/Conference — Submitted to ICLR 2026_

### Official Review · Reviewer_tC2p · 2025-10-24

**Soundness:** 4
**Presentation:** 4
**Contribution:** 3
**Rating:** 8
**Confidence:** 4

**Summary:**

The authors present a study on what happens when RL agents can decide how much compute to use, as measured by how many decisions they make, and as implemented by options of various durations. Their baseline comparison is to DQN, and they use the widely-used ALE set of environments. Results are positive, demonstrating that agents who can manage their own compute outperform agents who don’t.

1. **What is the specific question and/or problem tackled by the paper?**

    Put simply, “What happens when you let RL agents decide how to use their own compute?”

2. **Is the approach well motivated, including being well-placed in the literature?**

    Yes, very relevant work, cites relevant literature on RL including the options framework, DQN work, and related work on sample efficiency, etc.

3. **Does the paper support the claims? This includes determining if results, whether theoretical or empirical, are correct and if they are scientifically rigorous.**

    Claims are straightforward, that compute-aware RL agents may perform better under the same compute budgets as baseline agents. Results are straightforward to support the claim that compute-aware agents perform better in 75% of ALE environments, save for 3.

4. **What is the significance of the work? Does it contribute new knowledge and sufficient value to the community?**

    Yes, a straightforward hypothesis, simple and interpretable math, fits neatly into previous work and cites it well, is relevant to modern questions about compute efficiency, and tested on a huge number of environments with decent visualizations and analysis. This paper could use a journal version.

**Strengths:**

Very clear framing, extremely easy to understand, well formulated, positive results, cites relevant RL literature.

**Weaknesses:**

Could use more analysis as to the 25% negative results.

**Questions:**

I do wish there was slightly more commentary on the exploration/exploitation divide. It seems that temporally-extended options can make exploration more difficult, so some information about how much coverage of the ground-truth MDP agents make over time would be interesting.

A comparison to the baseline DQN with a frameskip of 10 (6Hz?) instead of 5 would be interesting.

Some analysis of the worse performing environments would be welcomed. Any insights there? Why do compute-aware agents struggle so much in these environments? Perhaps this ties into the exploration issue mentioned above.

---

> ### Author Response · Authors · 2025-11-20
>
> Thank you for your insightful comments.
>
> Addressing your main concern, it is difficult to pinpoint why Compute DQN performed much better in some games and had a decrease in others. High-level differences in games do not seem to be predictive as DemonAttack is similar to SpaceInvaders, and StarGunner is similar to ChopperCommand. We have videos available in the supplementary material if you are interested to contrast the games; we included only a subset due to upload restrictions in OpenReview. We could upload different videos to the supplementary material if that would be useful, as we will make them all available publicly when published.
>
> We should note that while improvements were sometimes hundreds of percent in HNS, the few cases where it performed worse is only 30% in HNS, we have added a raw HNS figure in the appendix to highlight this relatively small loss in performance.
>
> In section 4.1 we discuss possible reasons for the difference in performance between Compute DQN and DQN, such as improved credit assignment due to Compute DQN using an n-step update; exploration advantages that come from persisting with actions (see $\epsilon z$-Greedy; Dabney et al., 2021).
>
> > “It seems that temporally-extended options can make exploration more difficult, so some information about how much coverage of the ground-truth MDP agents make over time would be interesting.”
>
> As noted above, temporally-extended options have previously been proposed as a mechanism to improve exploration (see $\epsilon z$-Greedy; Dabney et al., 2021), rather than being a hindrance. As for providing coverage of the MDP, while this would be straightforward in tabular MDPs such as grid worlds (where such coverage is often shown) we don’t know how to do this on the arcade environments, which don't have enumerable states.
>
> **Our Questions:**
>
> > “I do wish there was slightly more commentary on the exploration/exploitation divide.”
>
> Could you please clarify what you mean by exploration/exploitation divide?
>
> **Request from the reviewer for additional experiments**
>
> We are trying to run additional experiments as suggested and hope to include them in a revision before the end of the response window.

---

> > ### Author Response · Authors · 2025-12-03
> >
> > We have added a fixed frame-skip ablation for DQN: comparing how Compute DQN fares compared to DQN with a fixed frame-skip aimed to roughly match Compute DQN’s average compute usage (4 decisions / second; or a frameskip of 15). The result under this compute constraint is that Compute DQN still outperforms DQN on 61% of the games. We believe this further demonstrates the value of allowing the agent to reason explicitly about its compute usage online. We have added figure 10 in the appendix to show the improvement in human normalised score (HNS), and figure 11 to show the raw HNS.

---

### Official Review · Reviewer_qkKs · 2025-10-30

**Soundness:** 3
**Presentation:** 3
**Contribution:** 3
**Rating:** 4
**Confidence:** 4

**Summary:**

This paper presents a method for augmenting DQN with action repeat options, showing that this reduces the number of actions executed per frame and improves human-normalized score (HNS) compared to the DQN baseline policy on about three-quarters of the Atari games.

**Strengths:**

- Clean formalization and direct application to any value based rl method
- Experimental results on all atari environments, showing less average decisions and higher HNS compared to existing baselines.
- Analysis of decision rate change intra episode is interesting and novel.

**Weaknesses:**

- Compute is only accrued at decision steps, which does not cover simulator overhead/option execution. Since the proposed agent might see more frames/episodes than the baseline agent during decision-making, this may confound the reported gains.
- The framework seems to heavily rely on the existence of temporally extended options. Given the prevalent use of action repeat / frame skip in the atari literature, this narrowly makes sense. In an arbitrary RL environment however its not obvious to me that these assumptions hold.
- Figure 8 shows HNS monotonically decrease wrt compute cost. This is surprising considering the overall claim of the paper that a nonzero c improves performance relative to DQN (c=0).
- No comparison with wider baselines like Rainbow, QR-DQN, IQN, or PPO or ablations on various durations of cleanup-target episodes is provided.
- The field of efficient reasoning, particularly in the Reasoning LLM context, is under discussed.

**Questions:**

1. It is not clear how the ratio of 40M decisions to 200M DQN frames is determined. Please report the average number of environmental frames sampled for each environment.
2. What differentiates the performance of the proposed method with c=0 from DQN?
3. How does compute savings hold when the cost of option execution / simulator cost is taken into account? An ablation here would improve the paper.
4. Run baselines on Rainbow/QR-DQN (with the fixed frame-skip ablation).
5. How do the author's position this paper in comparison to efficient llm reasoning line of work. Two papers which come to mind are:

Achille, Alessandro, and Stefano Soatto. "AI Agents as Universal Task Solvers." arXiv preprint arXiv:2510.12066 (2025).
Aggarwal, Pranjal, and Sean Welleck. "L1: Controlling how long a reasoning model thinks with reinforcement learning." arXiv preprint arXiv:2503.04697 (2025).

The first explicitly discusses the notion of the cost of time, which is analogous to this works notion of the cost of computation. The second can be seen as parameterizing cost as constraint rather then some lambda*t.  A discussion of this context, given the authors claim their method is, "a stepping stone toward agents that can reason about their own computational processes just as they reason about how they can bring about goals in the world," would strengthen the paper.

A discussion of this context along with clarifying my other concerns would move my score to an accept. I do not see any fundamental concerns with this paper.

---

> ### Author Response · Authors · 2025-11-20
> **Part 1/2**
>
> Thank you for your questions and feedback.
>
> > “Compute is only accrued at decision steps, which does not cover simulator overhead/option execution. Since the proposed agent might see more frames/episodes than the baseline agent during decision-making, this may confound the reported gains.”
>
> The simulator overhead and repeating actions (open-loop option execution) are negligible. First, running the emulator forward for one step requires much less computation than having the agent select an action (Bellemare et al., 2013; Mnih et al., 2015). Second, as our options are simply repeated actions, the agent skips processing inputs during these repeats making option execution nearly instantaneous. Analogous to skipping inputs during option execution, frame-skipping is used ontop of this and is a common way to significantly speed up the execution time of experiments (Machado et al., 2018), as an agent designer intervention. As the simulator overhead/option execution is negligible, most of the computational cost of running an experiment is in running the agent, which is also the only part of the computation the agent can control.
>
> The reviewer makes an astute observation that agents using less compute will cause the emulator to step more, resulting in more frames being computed and more episodes being completed under the training compute budget. While we could have restricted each agent to an environment-step budget, this would have focused the attention on sample efficiency rather than compute efficiency. In our setup, each agent sees the same total transitions (as some are option transitions with longer duration), the same number of decisions, and the same number of network updates.
>
> Even so, the degree to which Compute DQN benefitted from seeing more episodes than DQN can be observed by comparing figures one and two. If there was a benefit from more episodes seen by a method, then we would expect to see a correlation between fewer decisions per second indicating higher performance, as a lower decision rate requires the emulator to step more.
> We do not see sufficient evidence to support the claim that Compute DQN benefitted from the simulator running longer.
>
> We have added further discussion of this choice and its ramifications at the end of Section 3 and in the discussion of Figures one and two.
>
> > “The framework seems to heavily rely on the existence of temporally extended options. Given the prevalent use of action repeat / frame skip in the atari literature, this narrowly makes sense. In an arbitrary RL environment however its not obvious to me that these assumptions hold.”
>
> While action repeats do appear prevalently in the Atari literature, their applicability is not so narrow;
> action repeats have been advocated as an effective general-purpose exploration method (Dabney et al., 2021), named $\epsilon z$-Greedy, where experiments in a number of classic control problems and the ALE have showcased promising exploration benefits and improved performance on hard exploration games like MontazumaRevenge.
>
> With that said, the use of longer-duration options as the basis for reasoning about compute due to action frequency, can be more general. Options are not limited to action repeats, options may be arbitrary sequences of actions; performed open or closed loop.  Options can also be stochastic or contingent on events (Sutton et al., 1999). While not in scope for this work, options can even be learned (e.g., Bacon et al., 2017). And one would need to be careful that the computational cost of option execution is (i) accounted for if not negligible and (ii) much smaller than the computation cost of the agent's typical action selection. We mention this as a possibility in the discussion section.
>
>
> **References**
>
> Bellemare, M. G., Naddaf, Y., Veness, J., & Bowling, M. (2013). The arcade learning environment: An evaluation platform for general agents. Journal of Artificial Intelligence Research (JAIR).
>
> Mnih, V., Kavukcuoglu, K., Silver, D., Rusu, A. A., Veness, J., Bellemare, M. G., Graves, A., Riedmiller, M. A., Fidjeland, A. K., Ostrovski, G., Petersen, S., Beattie, C., Sadik, A., Antonoglou, I., King, H., Kumaran, D., Wierstra, D., Legg, S., & Hassabis, D. (2015). Human-level control through deep reinforcement learning. Nature.
>
> Machado, M. C., Bellemare, M. G., Talvitie, E., Veness, J., Hausknecht, M. J., & Bowling, M. (2018). Revisiting the arcade learning environment: Evaluation protocols and open problems for general agents. Journal of Artificial Intelligence Research (JAIR).
>
> Sutton, R. S., Precup, D., & Singh, S. (1999). Between mdps and semi-mdps: A framework for temporal abstraction in reinforcement learning. Artificial Intelligence.
>
> Dabney, W., Ostrovski, G., & Barreto, A. (2021). Temporally-extended ϵ-greedy exploration. International Conference on Learning Representations (ICLR).
>
> Bacon, P.-L., Harb, J., & Precup, D. (2017). The option-critic architecture. The AAAI conference on artificial intelligence.

---

> > ### Author Response · Authors · 2025-11-20
> > **Part 2/2**
> >
> > > “Figure 8 shows HNS monotonically decrease wrt compute cost. This is surprising considering the overall claim of the paper that a nonzero c improves performance relative to DQN (c=0).”
> >
> > The "overall claim of the paper" is not really that a nonzero c improves HNS performance relative to DQN. Rather it claims that a nonzero c significantly reduces the compute usage of the agent whilst not significantly harming HNS performance relative to DQN. We expected performance to be negatively impacted by our change, so, in light of your feedback we have expanded our discussion in section 4.1 to elaborate more on why Compute DQN improved in many games, and performed worse in a few.
> >
> > As for the monotonic decrease in Figure 8, we first note that Figure 8 only shows three games. Some games behave more like Pong or Breakout --- where a moderate compute cost has little effect on HNS, while significantly reducing the overall compute usage, and other games behave more like Asterix with a more substantial decrease in performance with a moderate compute cost).
> >
> > We also note Compute DQN with $c=0$ is not equivalent to DQN.
> > While both put all the emphasis on performance, Compute DQN is still choosing between the larger action/option set, which can have both potential advantages (credit assignment, exploration) and potential disadvantages (larger action space to learn, primacy bias focusing only on low decision rates from early training) in influencing performance and compute cost.
> >
> > > “It is not clear how the ratio of 40M decisions to 200M DQN frames is determined.  Please report the average number of environmental frames sampled for each environment.”
> >
> > When DQN is trained for 200 Million frames, DQN sees and selects actions on every kth frame instead of every frame, where we set _k_ to five for all games (as is common), resulting in 200 M frames / 5 skips = 40 M frames where the agent will compute an action selection. We have added a table in the appendix to show the average number of emulator frames for Compute DQN on each game.
> >
> > > "What differentiates the performance of the proposed method with c=0 from DQN?"
> >
> > See above.
> >
> > > "How does compute savings hold when the cost of option execution / simulator cost is taken into account? An ablation here would improve the paper."
> >
> > See above.
> > We are not sure what ablation the reviewer is referring to.
> >
> > > “How do the author's position this paper in comparison to efficient llm reasoning line of work.”
> >
> > There are indeed similarities to work on efficient LLM reasoning. Graves (2017) in even earlier work presented a "time penalty" into the training objective of an RNN to allow the algorithm to learn when to halt a recurrent computation and output an answer. Banino et al. (2021) learned a distribution over compute usage in recurrent computations while regularizing to a known distribution that preferred using less compute. Your more recent references to preprints (including a paper only available after the submission deadline) continue in this tradition. Achille and Soatto (2025), while doing a deeper dive into memory, training, and test-time compute, suggesting a "cost of time", which is indeed quite similar to our approach. In summary, our choice of how to incorporate the cost of computation is not particularly novel, nor did we see it that way. However, using a cost of computation within the RL problem is novel. Furthermore, issues that arise in sequential decision-making, where computational choices (e.g., lower decision rates) change how the agent interacts with the environment, are specific to RL. We have added a paragraph discussing this connection in the paper's Discussion section.
> >
> > The reviewer also points to Aggarwal and Welleck (2025) who suggest a token constraint on a reasoning model as a way to explicitly control the amount of test-time compute. We believe a trade-off weight is more natural for RL, as it allows the agent to identify and plan for circumstances when the prospect of future reward might warrant more compute. A per-decision compute constraint leads to a situation currently similar to traditional RL, where the agent designer's choice of model and algorithm effectively controls that per-decision constraint.
> >
> > **Our Questions:**
> >
> > > “No comparison with wider baselines like Rainbow, QR-DQN, IQN, or PPO or ablations on various durations of cleanup-target episodes is provided.”
> >
> > We are not certain what the reviewer means by "various durations of cleanup-target episodes"?
> >
> > **Request from the reviewer for additional experiments**
> >
> > We are trying to run additional experiments as suggested and hope to include them in a revision before the end of the response window.
> >
> > **References**
> >
> > Banino, A., Balaguer, J., & Blundell, C. (2021). PonderNet: Learning to ponder. AutoML Workshop @ International Conference on Machine Learning (ICML).
> >
> > Graves, A. (2017). Adaptive Computation Time for Recurrent Neural Networks. CoRR abs/1603.08983.

---

> > > ### Author Response · Authors · 2025-12-03
> > >
> > > We have included the results for DQN operating at a decision rate of 4Hz in the paper, please refer to our response to reviewer tC2p for an overview of these results. Additionally, we have added a table in the appendix to report the frames emitted from the emulator.
> > >
> > > We have initial results for Rainbow and Compute Rainbow. However, we could not use the default Rainbow n-step TD update of $n=3$ (they searched over {1,3,5}), and instead, we use $n=1$ for Compute Rainbow, as determining the appropriate way to incorporate an $n$-step update with options is not obvious. Otherwise, the Rainbow baseline was used in its default configuration as a baseline with Compute Rainbow adding the compute cost and different duration options. Our initial results show a similar pattern across the ALE: we see that Compute Rainbow significantly reduces the compute cost to a decision rate of 2.9Hz while performing better in more games than Rainbow. These results give further evidence that agents can reason about their compute, as with an improved agent design such as Rainbow, the agent has learned to further reduce its compute cost while maintaining a performance improvement across games.

---

### Official Review · Reviewer_b8ef · 2025-11-01

**Soundness:** 4
**Presentation:** 4
**Contribution:** 2
**Rating:** 6
**Confidence:** 4

**Summary:**

The paper introduces Compute DQN, an extension of Deep Q-Networks that enables agents to control and reason about their own computational effort. By integrating a compute cost into the reward function and allowing the agent to choose action durations, the model learns not only the policy but also the decision frequency. Evaluated on 46 Atari games, Compute DQN outperforms standard DQN in 75% of tasks while using much less computation on average. The agent adapts its decision rate dynamically, conserving compute in low-stakes moments and increasing it during critical gameplay phases. These findings demonstrate that agents can autonomously learn compute-efficient strategies without sacrificing performance, highlighting a path toward energy-efficient and adaptive reinforcement learners that optimize not only task outcomes but also their internal resource use.

**Strengths:**

1. The setup is clearly stated, and the overall flow is easy to follow. I enjoyed reading this paper a lot, and oftentimes I find answers to my questions/confusions lying just a few sentences away.

2. The idea of modeling deliberate control of computing budget with options under different frequencies is quite novel, and it turns out to be very effective.

3. The experiment evaluation is thorough and convincing.

**Weaknesses:**

If I have to say sth here, the only thing I would say is if the authors can show similar results on some larger tasks like VLA, LLM fine-tuning, it would make the work perfect.

**Questions:**

1. What do you think might be the reason that compute-DQN performs badly in some environments like Gopher and DemonAttack? Is it because reward/computing cost scale? Did you clip both of them by [-1, 1]? Have you tried a finer-grained option set to see if those game performances improve?

2. Under the computing budget + limited set of frequencies, could the authors envision how to analyze the suboptimality of the learned policy theoretically?

---

> ### Author Response · Authors · 2025-11-20
>
> Thank you for your thoughtful comments, especially on how we could improve our work.
>
> > “If I have to say something here, the only thing I would say is if the authors can show similar results on some larger tasks like VLA, LLM fine-tuning, it would make the work perfect.”
>
> While we are also interested in future work in language modeling tasks, we feel this is out of scope for the paper.
> Furthermore, we believe the arcade environments better showcase how agents are learning/reasoning about their compute.
> For example, we can visually see the agents adapt their compute, by acting less frequently during downtimes between rounds of play, or act more frequently when there are opportunities to improve the score.
> While we tried to show this in text and pictures in Section 4.3, the supplementary materials contain videos that better illustrate this adaptivity.
> We provided videos up to the allowed download limit and we could upload different videos to the supplementary material if that would be useful.
> We have videos for various seeds, games, and baselines; we will make all videos available publicly when published.
>
> > “What do you think might be the reason that compute-DQN performs badly in some environments like Gopher and DemonAttack?”
>
> It is difficult to pinpoint why Compute DQN performed much better in some games and had a minor decrease in others.
> High-level differences in games do not seem to be predictive as DemonAttack is similar to SpaceInvaders, and StarGunner is similar to ChopperCommand.
> We should note that while improvements were sometimes hundreds of percent in HNS, cases where it performed worse amounted to less than a 30% change in HNS relative to the baseline.
> To clarify the small performance drop, we have added a figure of raw HNS in the appendix.
>
> > "Did you clip both of them by [-1, 1]?"
>
> Yes.
> We follow the typical regime of using the sign of the score as the score varies wildly across games, this is the clipped reward you speak of, and the compute cost is added to the clipped reward.
>
> > “Have you tried a finer-grained option set to see if those game performances improve?”
>
> While we tried a few different sets of option lengths, this was the largest set of option lengths we ran.
> Adding more options can make the learning problem more challenging as there are more action values to learn.
> In fact, many games saw better performance with fewer available option lengths.
>
> > "Under the computing budget + limited set of frequencies, could the authors envision how to analyze the suboptimality of the learned policy theoretically?"
>
> Even without compute costs, given that we cannot analyze the suboptimality of learned policies in non-trivial environments (such as the ALE) this doesn't seem feasible.
> As Compute DQN includes the decision rate of DQN in its set of options, Compute DQN can always learn the same behavior as DQN, while the options give it the possibility of learning a more optimal tradeoff.

---

> > ### Comment · Reviewer_b8ef · 2025-11-26
> >
> > Thanks for your reply. I'll maintain my positive score on this.

---

### Official Review · Reviewer_qosa · 2025-11-03

**Soundness:** 1
**Presentation:** 1
**Contribution:** 2
**Rating:** 2
**Confidence:** 4

**Summary:**

The paper presents a method to expand the action space of an agent to decide to commit to taking the same action across a number of timesteps, so that no decisions need to be made for those timesteps. It also augments the reward to subtract a value for every time a decision is made. This then encourages the agent to make fewer decisions, thus reducing computation time. Results show that better performance can be obtained on the arcade learning environment, on average, while also reducing the number of decisions.

**Strengths:**

The paper’s methods are concise, and the results make sense. I believe the results can be easily replicated.

**Weaknesses:**

The paper positions itself about investigating algorithms that reason about their own computation. However, the method presented involves a straightforward modification to the reward given by the MDP. Therefore, this paper is related to reward shaping. However, relevant literature from this field is not mentioned and, therefore, the paper does not position itself within the relevant literature.

**Questions:**

What is meant by “reasoning” and can reward shaping be equated to enabling an algorithm to meta-reason, i.e. reason about its own computation?

---

> ### Author Response · Authors · 2025-11-20
>
> Thank you for your thoughtful comments.
>
> Reward shaping is typically the process where an agent designer provides additional rewards to guide (i.e., shape) the agent's behavior to learn the optimal behavior more quickly.
> However, additional control actions are not a typical component of reward shaping.
> In our work, we are not shaping the behavior but rather changing the agent's task to consider the cost of its computation (which could be considered "a straightforward modification of the reward)".
> We further provide the agent with compute actions (through specifying the option duration) that allow it to adapt its compute usage.
> Specifying the compute cost and providing actions to adapt the compute use does not easily fall under the umbrella of reward shaping alone.
> It is critical to our work that compute actions are provided and that feedback is given to the agent.
> We called our approach "reasoning" as the agent is explicitly representing the value of different option lengths as each option changes the decision rate and impacts the return under our explicit compute-performance trade-off; we have added this clarification on reasoning into the paper.
> We do not see how designer-provided reward shaping could be equated to such (meta-)reasoning.
>
> Nevertheless, from your comment we are interested in the future work on seeing whether reward shaping can improve agents that reason about their computation.
> In our work we specified the desired tradeoff between compute and performance in the reward function, which was fixed for the duration of learning.
> Instead, one could consider annealing the compute cost to the desired trade-off as a reward-shaping method, allowing the agent to first learn to play the game at a high-level before encouraging it to reduce its compute usage (or vice versa).
> That is an interesting suggestion that we have added as potential future work in the conclusion.

---

### Author Response · Authors · 2025-12-03
**Summary**

We want to thank the reviewers for their feedback and highlight the changes we have made to the paper during the rebuttal period in this comment.

We have added clarifications to the paper from the reviewer’s questions. For instance, we clarify that our method reasons about computation by explicitly representing the expected value over actions that change the agent’s computation. That the selection over compute-actions impacts the objective under our explicit compute-performance trade-off. And that reasoning is not limited by the options framework as options represent any sequence of actions. Furthermore, we have expanded on the untapped directions of future work both in the discussion section and the conclusion. These edits are visible in red text in the second revision.

We have run additional experiments as requested. Two reviewers wanted to compare DQN operating at the average decision rate learned by Compute DQN. We observe that our method still outperforms DQN at the lower decision rate on 61% of games, further demonstrating the value of allowing the agent to reason explicitly about its compute usage online. Additionally, we ran our method on Rainbow and obtained similar results to what we report with DQN, where Compute Rainbow (our method combined with Rainbow) learns an even lower decision rate of 2.9 Hz while having a higher performance across the ALE.

With our responses to all reviewers and the revisions to the paper, we believe we have addressed all the reviewers concerns.

---

### Meta-Review · Area_Chair_ZCWH · 2026-01-05

**Summary:**

The paper proposes a method for making RL agents account for the cost of policy inference while learning the policy. This is done by adding a "computation cost" to each action and introducing options that allow the agent to run for several timesteps in a row without paying any computation cost. In experiments done on Atari, the agents end up learning policies that are less compute-costly and also perform better than policies learned by standard DQN.

The reviewers had a number of questions/concerns about the experiment setup, the biggest of which were:
- Lack of mention of related work on reward shaping.
- Lack of empirical evidence that this approach works is applicable to large-model finetuning.
- The dependence of the paper's conclusions on the availability of options in the environment.
- Lack of analysis of why the method performs worse than standard DQN on some games.

In the metareviewer's opinion, the authors have provided reasonable responses to these concerns, except the last one, for which there is a straightforward hypothesis that the authors could have explored (see below). In particular, reviewer qosa, who criticized the lack of citations on reward shaping, didn't mention any specific methods from this field that, in their opinion, are relevant to the presented one and that the authors should have cited.

However, upon reading the submission, the metareviewer noticed a major methodological flaw that undermines the paper's conclusions and a more minor issue regarding related work, neither of which were mentioned by the reviewers.

The minor issue is the lack of citation and discussion of two major works that provide formalization for metareasoning about computation in the context of decision-making:

Wefald, Russell, "Principles of Metareasoning", Artificial Intelligence Journal, 1991

Lin, Kolobov, Kamar, Horvitz, "Metareasoning for Planning Under Uncertainty", AAAI-2015

The bigger issue, though, is that the paper's claim, that accounting for computation is the reason for achieving better performance, does not and cannot possibly follow from the paper's existing experiment setup. Namely, in the experiments, the authors take ALE games and

- Augment their action space with a set of action-repeat options, which have been previously shown to generally improve DQN's performance on ALE (the authors acknowledge this and cite the relevant literature).
- Make these options more "attractive" in terms of instantaneous reward, due to the added cost, than the choice of repeating the corresponding action N times.

In other words, in the experiments, DQN and the proposed method, Compute-DQN, solve MDPs that are different both in reward functions and in their action spaces. The authors acknowledge this in the rebuttal to reviewer qkKs, who asked about the equivalence of DQN and Compute-DQN under c=0. **The critical aspect here, though, is that DQN is given the MDPs whose action space without the action-repeat options -- the action space that is empirically known to be harder for DQN in the context of ALE -- than the MDPs that Compute-DQN is given.** As a result, we can't be sure why Compute-DQN, on average, finds better policies on ALE: whether it's because of the extended action space or the reward modification induced by the cost of computation. In fact, given DQN's previously established better performance on ALE in the presence of action-repeats, there is a strong reason to believe it's the former.

To *potentially* address this, two things need to be done:

- Do experiments comparing DQN with ALE's standard action space vs. DQN with the same action space you are giving Compute-DQN, but without any cost of compute. If the latter wins, at least some of Compute-DQN's current performance lift is attributable to the augmented action space.

- Do experiments on 2-3 other environments. As implied above, ALE with options - which the presented method critically relies on -is already known to be biased towards being more convenient for DQN.

Note, however:
- The need to augment the original MDP with options will remain a confounding factor in any other environment. A major motivation behind using options is "helping" any given RL algorithm by shortening the MDP's effective horizon, so the set of options for a given problem is generally carefully constructed to contain helpful ones, regardless of the effect of the cost of computation.

- There is no clear intuition why the cost of computation *in general* should help choose better policies. Generally, it may very well make options that are worse under the original reward look more attractive. We may actually be seeing this phenomenon in Figure 8. Therefore, it's unclear how the effect of computation cost on the selection of better (or worse) policies can *ever* be anything other than an experiment fluke, as it almost surely is in the ALE experiments.

To summarize, in the current form, the submission's claims are methodologically compromised, so the submission isn't ready for publication.

**Reviewer Concerns:**

The reviewers had a number of questions/concerns about the experiment setup, the biggest of which were:
- Lack of mention of related work on reward shaping.
- Lack of empirical evidence that this approach works is applicable to large-model finetuning.
- The dependence of the paper's conclusions on the availability of options in the environment.
- Lack of analysis of why the method performs worse than standard DQN on some games.

In the metareviewer's opinion, the authors have provided reasonable responses to these concerns, except the last one, for which there is a straightforward hypothesis that the authors could have explored (see below). In particular, reviewer qosa, who criticized the lack of citations on reward shaping, didn't mention any specific methods from this field that, in their opinion, are relevant to the presented one and that the authors should have cited.

**Reviewer Scores:**

Reviewer qkKs promised to consider changing their original score of 4 to an "accept" score if their concerns were addressed, which they were in the metareviewer's opinion. The other reviewers' scores would likely remain the same.

---

### Decision · Program_Chairs · 2026-01-26

Reject